## RESEARCH ARTICLE

# Neurotranscriptomic profiling of deformed wing virus-infected honey bee foragers (*Apis mellifera*) with different cognitive abilities

Simon E. Loughran[1,*], Lauren Dingle[1], Alan S. Bowman[1] and Fabio Manfredini[1,2,*]

## ABSTRACT

Honey bees (*Apis mellifera*) provide important ecosystem services to both natural and human-managed environments, but are increasingly threatened by a variety of pathogens, the most common of which is deformed wing virus (DWV). DWV is known to replicate in the honey bee brain and has been documented as both improving and impairing olfactory learning and memory. We examined the transcriptomic response of the honey bee mushroom bodies – an area of the insect brain associated with higher cognitive functions – in bees with naturally occurring DWV infections, which varied in their ability to perform an associative learning task. RNA-sequencing analysis detected increased expression of genes involved in the immune response, including important antimicrobial peptides such as hymenoptaecin, apidaecin, and abaecin, and the downregulation of lysozyme, prophenoloxidase, and other genes associated with responses to a range of stressors. Additionally, gene ontology enrichment analysis revealed overrepresentation of key biological processes that form part of the immune response. We also noted significant differential expression of long non-coding RNAs (lncRNAs) presumed to be acting in a regulatory manner, and used these lncRNAs to construct gene regulatory networks. Strikingly, in contrast to previous studies on bees with artificially induced infections that have examined viral loads in the abdomen and non-specific areas of the brain, no correlation between DWV load in the mushroom bodies and cognitive function was noted. This highlights the complexity of host-pathogen interactions in honey bee neural tissues and the benefits of a spatially refined approach to brain transcriptomics in naturally occurring infections.

KEY WORDS: DWV, Mushroom bodies, Associative learning, Gene regulatory networks, Transcriptomics, Honey bees

## INTRODUCTION

Insect pollinators, like many bee species, provide important ecosystem services to both natural and human-managed environments, favouring the transfer of pollen from male to female flower parts with their incessant foraging activity. Foraging requires not only the high energy expenditure associated with flight – a metabolically intensive behaviour (Rothe and Nachtigall, 1989;

[1]School of Biological Sciences, University of Aberdeen, AB24 3UL Aberdeen, UK. [2]Department of Food, Environmental and Nutritional Sciences (DeFENS), University of Milan, 20133 Milan, Italy.

*Authors for correspondence (s.loughran.20@abdn.ac.uk; fabio.manfredini@unimi.it)

S.E.L., 0009-0003-1924-0397; F.M., 0000-0002-9134-3994

Schippers et al., 2010) – but also a sophisticated suite of cognitive skills, including the ability to orient, navigate, memorise and learn (Bullinger et al., 2023; Doussot et al., 2024). It is well established that a range of stressors can take a significant toll on the foraging behaviour of many bee species (Klein et al., 2017), either by affecting their energy levels and metabolism (Bordier et al., 2017), by altering their floral preferences (Koch et al., 2017), or by directly compromising their cognitive abilities (Gómez-Moracho et al., 2017; Klein et al., 2017). For example, land use and climate change, and resultant habitat loss and fragmentation, are associated with a reduction in the richness, diversity, and abundance of wild bees, smaller body sizes, and a decline in pollination services (Grab et al., 2019; Kammerer et al., 2021). In honey bees, exposure to sublethal doses of many insecticides can affect learning and memory (Belzunces et al., 2012; Siviter et al., 2018), including specific effects on the mushroom bodies (Palmer et al., 2013; Peng and Yang, 2016), while parasites and pathogens can impact multiple traits, including foraging activity and age of onset of foraging, social interactions, learning and memory, and motor behaviour (Gómez-Moracho et al., 2017). Honey bee colonies are typically simultaneously exposed to multiple stressors, resulting in complex interactions and antagonistic and synergistic effects (Straub et al., 2020; French et al., 2024), and a compromised immune system, which can exacerbate the impact of each stressor (El-Seedi et al., 2022). It is noteworthy that many of the observed effects are similar for different stressors, suggesting similar mechanisms of action within target tissues such as the brain. An integrated honey bee stress pathway has been hypothesised (Even, et al., 2012); however, this model focuses on a general stress response to acute and short-term stressors, and the mechanisms by which key stressors alter brain functions over a longer period of time to produce the observed behavioural changes remain largely uncharacterised.

The majority of viruses infecting honey bees are single-stranded RNA viruses, icosahedral in shape, 30 nm in size, and do not cause symptoms in individuals that are infected (de Miranda et al., 2013; McMenamin and Flenniken, 2018; Beaurepaire et al., 2020). Both horizontal and vertical transmission have been observed, and infection and presence of disease can vary according to life stage and season (de Miranda et al., 2013; Brutscher et al., 2016; Beaurepaire et al., 2020). Of these, deformed wing virus (DWV) is the most widespread and is recognised as a major honey bee stressor and contributor to colony demise (Wilfert et al., 2016; Kevill et al., 2019). When transmitted by the ectoparasite *Varroa destructor*, which feeds on the fat bodies and haemolymph of honey bee pre-imaginal stages (Santillán-Galicia et al., 2010; Martin et al., 2012; Ramsey et al., 2019), DWV reaches extremely high levels, capable of producing abnormalities at the morphological and physiological levels such as deformed wings and bloated abdomens, and can also lead to early mortality (Koziy et al., 2019). However, DWV can also be transmitted orally via trophallaxis or shared resources (de Miranda and Genersch, 2010), venereally, and vertically (de Miranda and Fries, 2008). Oral transmission typically results

in asymptomatic infections or milder symptoms, such as an early transition from nursing to foraging and impaired learning (Traniello et al., 2020). Disruption of foraging behaviour itself, such as a reduction in flight distance and duration, can also occur (Wells et al., 2016). It is generally assumed that asymptomatic infections are better tolerated by honey bee colonies, but preliminary evidence has shown that even asymptomatic infections could have a significant toll on colony fitness and survival (Benaets et al., 2017; Penn et al., 2022), warranting further investigation. Furthermore, the mechanisms by which DWV produces the reported behavioural changes are largely unknown, and their study could contribute to the broader field of the neurobiology of insects.

DWV has been detected in several regions of the honey bee brain (or neuropils) such as the optic lobes, antennal lobes and mushroom bodies (Shah et al., 2009). The presence of the virus in these regions strongly supports the hypothesis that infection may impact various sensory activities that are essential for a bee forager to perform effectively, such as visually identifying rewarding flowers and detecting the scent or sugar concentration of nectar (Iqbal and Mueller, 2007). DWV localisation in mushroom bodies is particularly noteworthy since this region of the insect brain is home to important cognitive functions such as learning and memory, which are fundamental for foragers to recall which flowers they have visited, and which of these are most rewarding. Intriguingly, previous research has shown that not only the genomic strand but also the replicating strand of DWV can be detected in the mushroom bodies of infected bees (Shah et al., 2009), supporting the hypothesis that this virus may interfere directly with neural functions associated with foraging behaviour.

Many studies have examined the impact of viral infections on abdomen or whole body transcriptomes of honey bees (Doublet et al., 2017), and analysis of brain transcriptomes has become more common in recent years (Zhao and Liu, 2022). Pizzorno et al. (2021) have recently specifically explored differentially expressed genes in isolated brain tissue from bees exposed to abdominal injections of a DWV inoculum. The study reported the overexpression of genes associated with immune response and, given the documented presence of DWV in the mushroom bodies, it can be hypothesised that immune-related genes should also be present more specifically in this area.

The link between DWV and honey bee behaviour has been explored only partially. Previous research indicates that DWV-infected honey bees perform poorly in classic learning and memory tasks, as evaluated with the proboscis extension reflex (PER) (Iqbal and Mueller, 2007; Chen et al., 2021a,b), a typical association paradigm that is used to assess cognitive abilities in a fully controlled experimental setup. However, these studies typically introduce the virus into adult worker bees using artificial methods – either by injection with a syringe to mimic natural *Varroa*-mediated transmission, or by feeding to replicate natural oral routes of infection. As such, some important steps associated with the pathogenicity and tissue tropism of DWV in the pre-imaginal stages of the host are bypassed with these approaches. This is a key moment in the interaction between host and pathogen, since the majority of natural infections begin precisely at the pre-imaginal stage and could have significant consequences for later developmental stages. Most importantly, the honey bee neural system is still not fully formed at the pre-imaginal stages and only reaches full maturity in adult bees (Farris et al., 1999). Intriguingly, a recent study by Szymański et al. (2024) investigated the effect on cognitive abilities of natural DWV infections in the mushroom bodies of honey bee foragers and, in contrast to what has been reported in previous

research, detected a subtle positive effect of the virus on reversal learning – but no effect in a simple associative learning task. This demonstrates a need for a focus on natural DWV infections, in order to provide a more comprehensive account of the interplay between this virus – in particular when detectable in the brain – and honey bee cognitive abilities, highlighting potential trade-offs between brain functions and immune responses that are triggered by viral infections (Pizzorno et al., 2021; Zanni et al., 2023).

Here, we focused on the analysis of honey bee foragers displaying natural infections with DWV loads ranging from low to high (but still lacking the deformities that would render a worker bee unviable as a forager). We assessed the cognitive abilities of these bees using a simple associative PER assessment combined with intermediate-memory retention, and selected bees that performed very well in the task (Good Learners) and bees that failed completely (Poor Learners). We then quantified with real-time quantitative PCR (RT-qPCR) the exact number of DWV copies (or genome equivalents) in the mushroom bodies of bees belonging to the two groups and performed a transcriptomic profiling of the mushroom bodies of these bees, with the aim of detecting the most-significant gene expression profiles associated with cognitive performance and DWV infection (Fig. 1).

## RESULTS

### Gene expression patterns in honey bee mushroom bodies

We initially performed a series of analyses to explore the overall patterns of gene expression in the 33 mushroom body samples, and to enquire whether there was any factor in particular responsible for major differences in transcriptomic profiles. Principal component analyses (PCAs) showed that samples clearly separated on both PC1 and PC2 according to colony of origin (Fig. 2A; see also additional PCAs in Fig. S6). There was little clustering due to behavioural test score, suggesting no clear effect of learning performance on global gene expression. A scree plot is included in Fig. S1.

PCA revealed that colony is an important factor driving global patterns of gene expression in this study (Fig. 2A). It is of note that Colony A included four samples with very high viral loads (>9 $\log_{10}$), compared to only two such samples from Colony B, reflecting the typically higher prevalence of DWV (Table S4) in the apiary incorporating Colony A (Bradford et al., 2017). In addition, a heatmap analysis suggested greater clustering for Colony A compared to Colony B (Fig. 2B), and this pattern was confirmed with cluster analysis: Colony A samples exhibited significantly tighter transcriptomic clustering relative to Colony B, as indicated by a higher average silhouette width (0.126 versus 0.054, $P$=0.0003488, Wilcoxon test) and a lower – though not statistically significant – mean within-group distance in PCA space (12.958 versus 14.252, $P$=0.1061, Wilcoxon test), calculated using the first three principal components, which account for 49.7% of total variance. These results confirm a greater degree of transcriptomic similarity among Colony A bees.

### Transcriptomic profiling of different groups of learners

The main goal of this study was to assess the impact of mushroom body viral infection on learning performance and to explore whether this is reflected at the transcriptomic level: in a series of analyses, we therefore created subsets of good and poor learners from both colonies, and within each subset we compared gene expression levels of individuals carrying high viral loads to those carrying low viral loads. These more-nuanced analyses gave us greater power to identify the subtle effects of viral infection on learning performance. The analysis on Poor Learners (16 individuals in total) revealed a

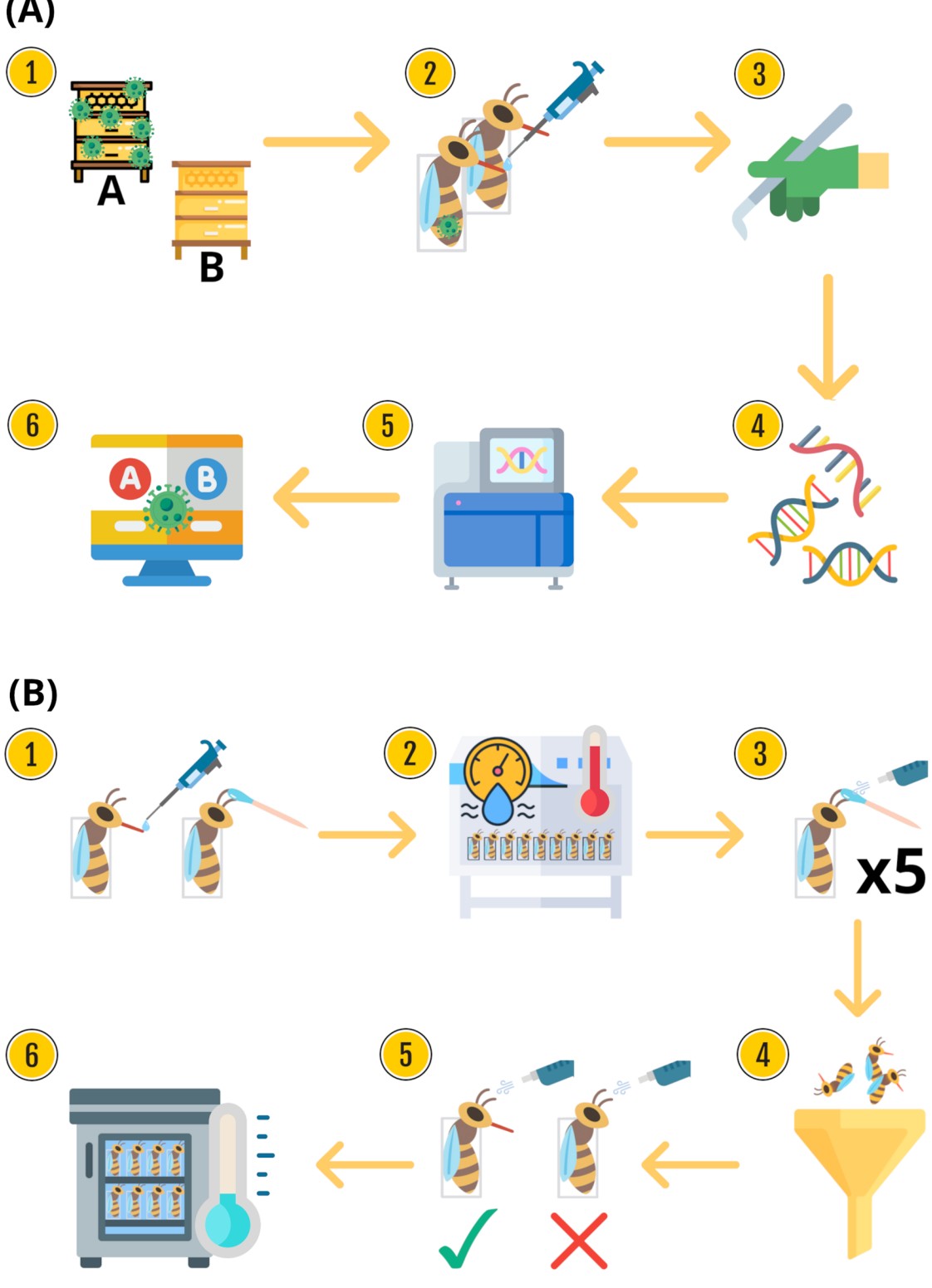

**Fig. 1. Schematics of experimental methods.** (A) Broad overview of the project's methods. (1) Sample collection. Collection of samples from Colony A (minimal *Varroa* treatment, higher DWV, *n*=16) and Colony B (standard *Varroa* treatments, lower DWV, *n*=17). (2) PER conditioning. Classification of individuals as Good Learners or Poor Learners (see Fig. S2 for details). (3) Mushroom-body dissection. Dissection of mushroom bodies followed by RNA extraction. (4) Quantification of DWV loads. RT-qPCR assay to assign High and Low DWV loads to individuals. (5) RNA-seq. Illumina libraries prepared and sequenced; reads processed and gene counts determined. (6) Gene expression analyses. Differential expression (DESeq2), functional enrichment (GO, KEGG), and gene regulatory networks (GRNs) on *n*=33 mushroom-body samples. (B) PER conditioning assays. (1) Initial feed and water desensitisation. Harnessed bees given 3–5 µl 30% (w/w) sucrose; antennae touched with water. (2) Acclimatisation. 1 h in darkness (lab cabinet), 22–25°C, 35–40% relative humidity. (3) Conditioning. Five citral presentations (5 µl 95% v/v on filter paper in 20 ml syringe) each reinforced with 30% sucrose. (4) Filter. Discard bees showing PER to first odour presentation or no PER to sucrose. (5) Memory tests. Three trials same day+three trials next day. Group assignment: Good Learners=PER in all six tests; Poor Learners=no PER in any test. (6) Storage. Freezing at −80°C for later molecular work.

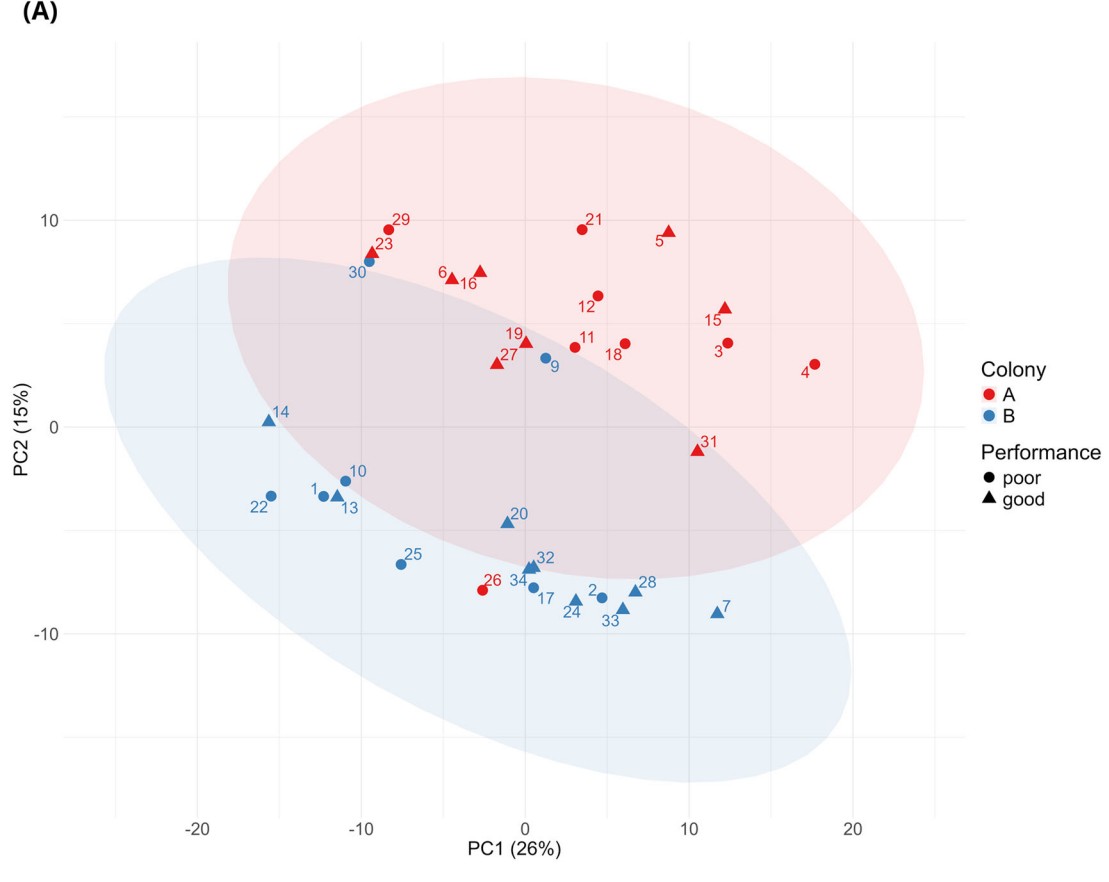

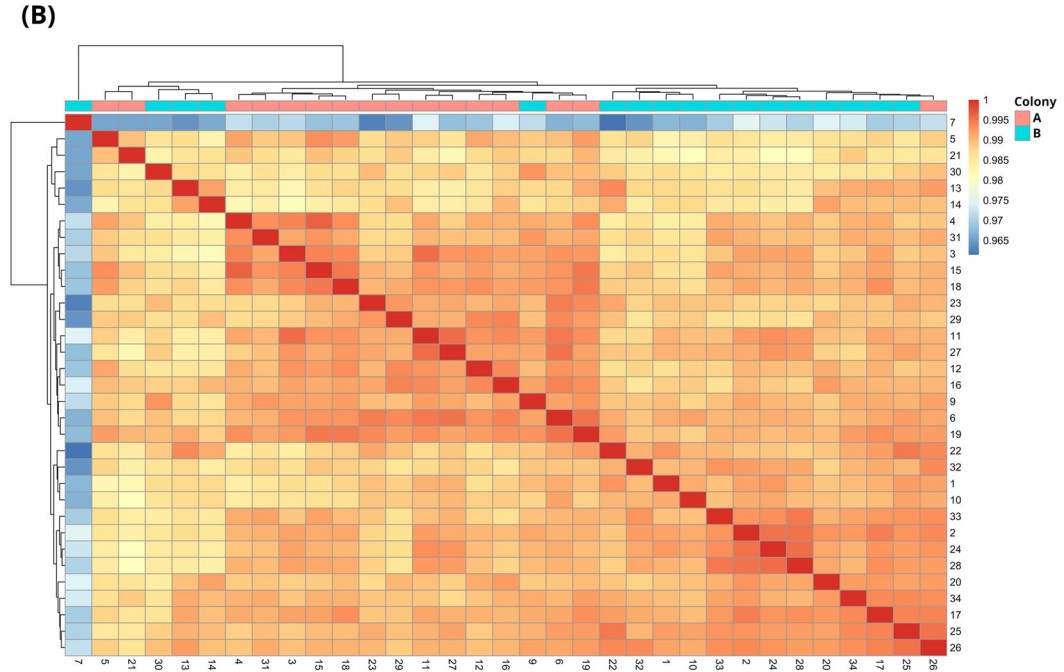

**Fig. 2. Clustering.** (A) Principal component analysis (PCA) showing clustering of the 33 mushroom body samples according to colony of origin, and absence of clustering according to performance in behavioural tests. The samples represent independent biological replicates collected from two separate colonies. PCA was performed using the normalised gene expression counts for the 500 most-variable genes expressed in the samples. Each point represents one of the 33 samples and is colour coded to indicate colony of origin and shaped to indicate learning performance. 95% confidence ellipses are superimposed for the colony of origin. PCs 1 and 2 together account for 41% of variation. (B) Heatmap showing hierarchical clustering of mushroom body gene profiles across 33 honey bee mushroom body samples. Each cell shows the pairwise similarity between two samples, calculated using Pearson correlation on variance-stabilised expression values of 12,332 genes. The samples are colour coded according to colony or origin (A or B) and represent biological replicates from individual mushroom bee bodies. Each sample was sequenced once. Colony A (*n*=16); colony B (*n*=17).

set of 58 genes that were significantly differentially expressed (Benjamini-Hochberg-adjusted $P<0.05$). Of these, 15 genes were expressed with an absolute $\log_2$ fold change of one or more ($|\log_2 FC|>1$) (Fig. 3A, Table 3): 80% of these genes were expressed at lower levels in Poor Learners when viral infection was high.

Gene ontology (GO) analysis revealed enrichment (Bonferroni-adjusted $P<0.01$) of key groups: GO:0007601/visual perception

**(A)**

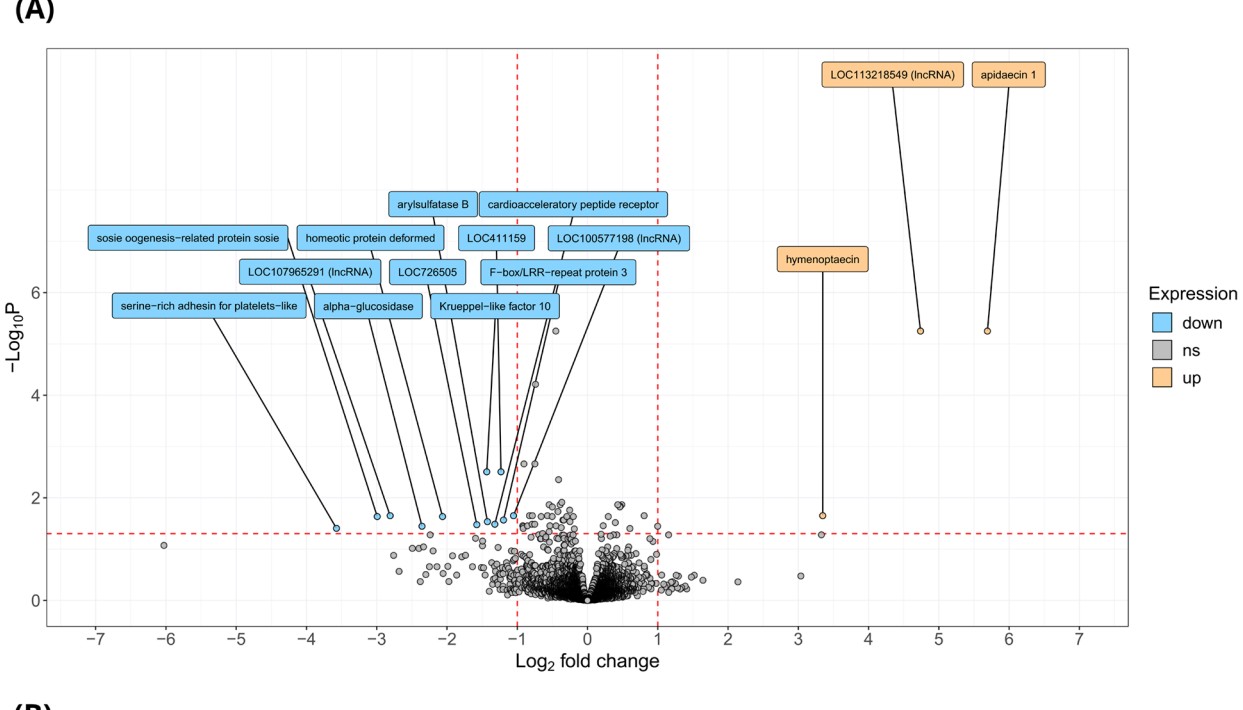

**(B)**

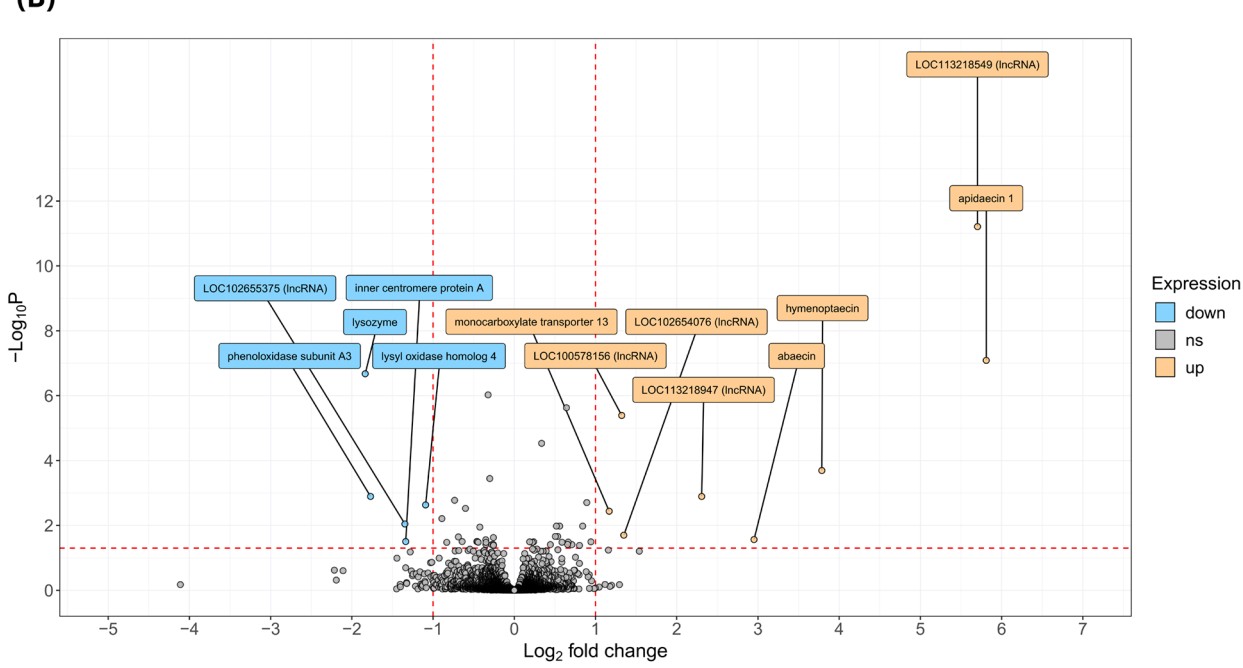

**Fig. 3. Volcano plots showing differentially expressed genes in mushroom bodies of DWV-infected Poor Learners and bees from Colony A.** Significant transcriptomic differences were revealed by carrying out differential expression analysis comparing Poor Learners with high viral loads ($n$=6) and Poor Learners with low DWV loads ($n$=10) (A) and comparing bees from Colony A with high viral loads ($n$=6) and those with low DWV loads ($n$=10) (B). Each point in each of the plots represents a single gene. Genes with an absolute log2 fold change of 1 or more ($|\log_2 FC|>1$; vertical red dotted lines) and a Benjamini-Hochberg-adjusted $P<0.05$, (horizontal red dotted line) are considered significantly differentially expressed. Genes in orange in A are more highly expressed in Poor Learners with high viral loads ($\geq \log_{10}$ 5 DWV genome equivalents), while genes in blue are more highly expressed in Poor Learners with low viral loads ($\leq \log_{10}$ 4 DWV genome equivalents) and are therefore expressed at lower levels in Poor Learners with high viral loads. The same colour coding is used in B but this time with reference to all Colony A samples. RNA was extracted from the mushroom bodies of individual bees and sequenced. Each sample represents an independent biological replicate. DESeq2 was used to assess differential expression using Wald tests, and $P$-values were adjusted for multiple testing using the Benjamini-Hochberg method. All tests were two-tailed.

(LOC726228, *Lop2*, *Lop1*), GO:0045087/innate immune response, and GO:0002376/immune system process (*Apid1*, LOC406142, LOC406144); there was also enrichment of GO:0004930 and GO:0007186, G protein-coupled receptor signaling pathways. Kyoto Encyclopedia of Genes and Genomes (KEGG) pathway analysis revealed pathways associated with phototransduction and neuroactive ligand-receptor interaction, but neither of these met our chosen significance threshold (Bonferroni-adjusted $P$=0.05) (Table S1).

When comparing bees with high versus low viral load within the subset of Good Learners (17 individuals in total), no genes were identified as significantly differentially expressed at our chosen threshold (Benjamini-Hochberg-adjusted $P$≥0.05).

### Transcriptomic profiling in response to viral infection

In a third set of analyses, we specifically focused on exploring the effect of harbouring a natural DWV infection on the transcriptomic profile of mushroom bodies. The tighter clustering of Colony A samples (Fig. 2) suggested a more consistent response to DWV infection and provided a strong biological rationale for proceeding in our analyses with this colony alone (hence excluding Colony B samples). By doing so we minimised the effects of confounding environmental and genetic factors, and with less noise we were better placed to detect biologically significant gene expression patterns. Supporting the decision to focus on Colony A bees, DESeq2 analysis of Colony B alone yielded no genes that were identified as significantly differentially expressed at our chosen threshold (Benjamini-Hochberg-adjusted $P$=0.05), and analysis of all 33 samples combined (Colony A and B) identified only two genes below this threshold: LOC413942, an acyl-CoA synthetase short-chain family member, and LOC102655661, ecdysone-induced protein 74EF-like.

Focusing exclusively on Colony A (16 individuals in total), we next examined how viral load influenced gene expression within the mushroom bodies, while disregarding learning performance. This analysis revealed a set of 50 genes that were significantly differentially expressed (Benjamini-Hochberg-adjusted $P$<0.05). Of these, 13 genes were expressed with an absolute $\log_2$ fold change of one or more ($|\log_2 FC|$>1) (Fig. 3B, Table 4). GO enrichment analysis revealed that key biological processes were overrepresented among these genes (Bonferroni-adjusted $P$<0.001): GO:0042742/ defence response to bacterium, GO:0002376/immune system process, and GO:0045087/innate immune response. The same three genes were significant in each group: *Apid1*, LOC406142, and LOC406144. KEGG pathway analysis identified enrichment in pathways associated with phototransduction and neuroactive ligand-receptor interaction, but neither of these results met our chosen significance threshold (Bonferroni-adjusted $P$=0.05) following the Bonferroni correction (see Table S2).

### Correlation between gene counts and viral loads

To explore at a finer scale the possible interaction between DWV particles in the mushroom bodies and brain gene expression, we performed correlations analyses (Spearman's rank correlation coefficient) in which we plotted viral genome equivalents (expressed as the $\log_{10}$ transformation of total viral counts) against the levels of expression of individual genes (expressed as the $\log_{10}$ transformation of normalised read counts). These analyses were performed on all differentially expressed genes contained in the two datasets of interest, Poor Learners and Colony A: both datasets revealed some interesting patterns. Four genes showed a particularly strong correlation in the Poor Learners dataset (rho≥0.75, $P$<0.001):

LOC724642, LOC411159, LOC726505, and LOC107965291. Fig. 4 shows plots for LOC724642/F-box/LRR-repeat protein 3 (panel A) and LOC107965291, a non-coding RNA (panel B). Three genes showed a particularly strong correlation in the Colony A dataset (rho≥0.74, $P$<0.001): LOC724899, LOC408544, and LOC113218947. Fig. 4 also shows plots for LOC724899/ Lysozyme (panel C) and LOC113218947, a non-coding RNA (panel D). Figs S2–S5 show correlation plots for all genes listed in Tables 3 and 4.

### Gene regulatory networks

The set of gene expression analyses that we performed highlighted the presence of several long non-coding RNAs (lncRNAs) in our list of differentially expressed genes. We decided therefore to explore the potential role of these molecules as regulators of gene expression by building gene regulatory networks (GRNs) for both datasets. GRN analysis of the Colony A dataset identified potential targets for four lncRNAs: LOC102654076, LOC102655375, LOC113218549 and LOC113218947; analysis of the Poor Learners dataset identified potential targets for two lncRNAs: LOC113218549, and LOC107965291. Table 5 lists the proposed target of lncRNA LOC113218549, which is common to both outputs (the other lncRNAs and their targets are shown in Tables S2–S4). Both the Colony A and Poor Learner GRNs conformed to a scale-free topology (Kolmogorov–Smirnov $P$≥0.99999; α≈2.6–2.7), meaning that their degree distributions follow a power law. In such networks, most nodes have few connections, while a small number of hubs have many. This property is characteristic of biological networks and supports the biological plausibility of the identified lncRNA hubs and regulator–target interactions.

### DISCUSSION

In this study, we investigated the impact of naturally occurring DWV infections on gene expression in honey bee foragers within a particular region of the brain – the mushroom bodies. Our initial analyses revealed that global gene expression patterns were primarily influenced by colony of origin, with Colony A showing distinct clustering patterns, higher DWV prevalence, and elevated viral loads, providing us with increased biological and statistical power to detect DWV-driven differential gene expression. A distinct set of differentially expressed genes was associated with higher DWV loads in this analysis, including several immune-related genes and lncRNAs. A similar analysis on Poor Learners revealed that high viral loads were linked to the differential expression of genes involved in immune response and sensory processing, including upregulation of antimicrobial peptides such as *apidaecin* and *hymenoptaecin*, and downregulation of genes related to neural signaling and metabolism. In contrast, no significant gene expression changes were observed in Good Learners when comparing high and low viral loads. Perhaps the limited sample size (17 bees when considering Good Learners alone) could explain this outcome, and a larger set of samples would increase the statistical power to detect significant differences; it must be noted though that the sample size we used is in line with previous studies on a similar topic (e.g. Pizzorno et al., 2021). Additionally, across both Poor Learners and Colony A bees, we found strong correlations between DWV load and the expression of specific genes, including immune effectors such as *apidaecin* and *argonaute-2*, and several lncRNAs. These genes, in particular, were not only substantially upregulated, but also tightly connected to a series of genes of interest (especially immune genes) in our GRN analysis, highlighting their potential function as regulators of the insect

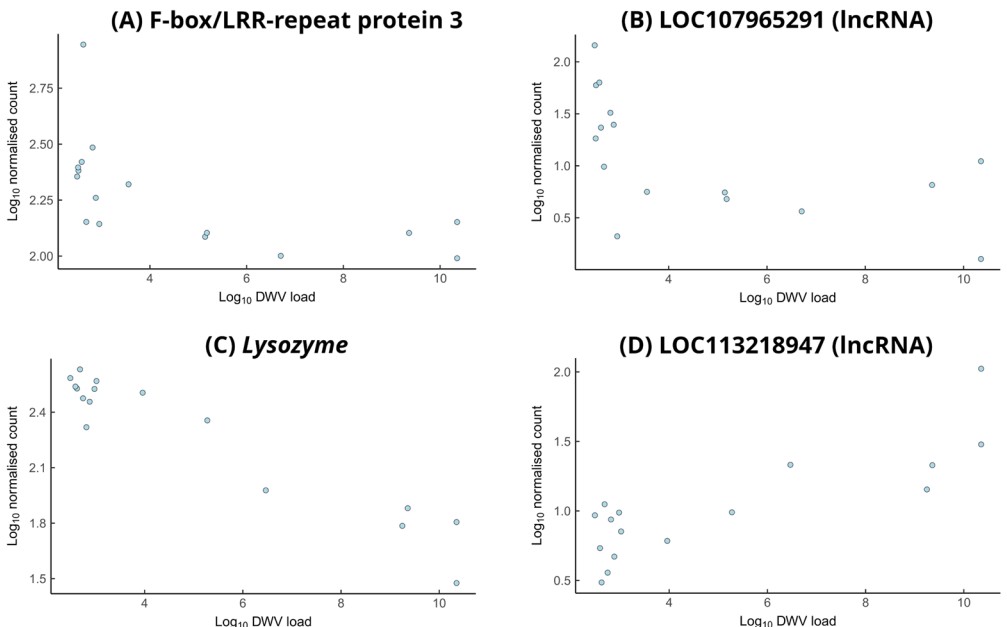

**Fig. 4. Genes showing strong correlations between expression and DWV load in mushroom bodies of DWV-infected bees.** Each panel shows an individual gene with significantly correlated expression relative to DWV genome equivalents in either Poor Learners or bees from Colony A. All correlations were calculated using Spearman's rank correlation (two-tailed). RNA was extracted from the mushroom bodies of individual bees and sequenced. Each point represents an individual biological replicate. Expression values are variance-stabilised counts. (A) LOC724642 (F-box/LRR-repeat protein 3): expression negatively correlates with DWV load in Poor Learners (rho=−0.79, $n$=16, $P$=2.57×10$^{-4}$). (B) LOC107965291 (long non-coding RNA): expression negatively correlates with DWV load in Poor Learners (rho=−0.75, $n$=16, $P$=8.08×10$^{-4}$). (C) LOC724899 (*Lysozyme*): expression negatively correlates with DWV load in bees from Colony A (rho=−0.83, $n$=16, $P$=6.18×10$^{-5}$). (D) LOC113218947 (long non-coding RNA): expression positively correlates with DWV load in bees from Colony A (rho=0.74, $n$=16, $P$=9.7×10$^{-4}$).

immune response and their role in modulating the honey bee brain's response to viral infection.

Notably, we observed no obvious transcriptomic signature in the mushroom bodies associated with learning performance. This appears counterintuitive, given frequent reports linking learning performance with mushroom body activity and gene expression. (Giurfa, 2003; Szyszka et al., 2008; Strube-Bloss et al., 2011; Fahad Raza et al., 2022). However, establishing a link between gene expression and behaviours that are highly transient is always a challenging endeavour. For example, the most suitable timing of capture and dissection of bees involved in learning tasks will vary according to needs (Clayton, 2000; Rittschof and Hughes, 2018). Rittschof (2017) cautions that although patterns of expression may indicate a tendency towards certain behaviours, this does not always translate to actual performance of these behaviours. It is also possible that other brain regions play a larger role – or act synergistically with the mushroom bodies – in regulating the response to an associative learning task. For example, the optic lobes or antennal lobes, which are more directly linked to the specific cues (visual and olfactory, respectively) presented during a PER assay. Szymański et al. (2024), who also investigated learning performance of bees carrying natural DWV infections, did not see any change in gene expression with simple associative learning tasks, but did note changes with a more complex reversal learning assay. Notably, Szymański et al. (2024) targeted a specific group of GABA-related genes with a real-time quantitative PCR (qPCR) approach, while we assessed the whole transcriptome of the honey bee with RNA sequencing (RNA-seq). Interestingly, there were no GABA-related genes in our lists of differentially expressed genes. This could be due to differences in the complexity of the cognitive assay that we adopted, as discussed above, or due to differences in sensitivity between the real-time qPCR assay and the RNA-seq

approach. In line with these considerations, we cannot exclude that the presence of DWV infections in the mushroom bodies might result in the modification of other types of learning, for example social learning, that is widespread in honey bees and other social insects, relies on complex systems of communications and has a transcriptomic basis (e.g. Veiner et al., 2022; Manfredini et al., 2023). As a matter of fact, DWV and other honey bee viruses have been shown to interfere with chemical recognition and odour perception (e.g. Geffre et al., 2020; Silva et al., 2025), and it will be interesting in the future, for example, to investigate whether such viruses have any effect on the honey bee waggle dance communication.

The most interesting discovery from our study was the detection of a clear transcriptomic signature associated with DWV infection in the mushroom bodies, evident in both the Poor Learner and Colony A datasets. DWV is known to actively replicate within the brain, including within the mushroom bodies (Shah et al., 2009), and we show here that immune genes are significantly expressed in this region of the honey bee brain. In both datasets, we observed upregulation of the antimicrobial peptides (AMPs) *apidaecin* and *hymenoptaecin*, while *abaecin* was upregulated in the Colony A dataset alone, suggesting possible variation in immune gene activation according to genetic and/or environmental background of individual colonies. Consistent with these findings, GO analysis revealed significant overrepresentation of immune-related categories in both groups, supporting the proposition that immune activation takes place within the mushroom bodies and that the insect brain therefore is not such an 'immune-privileged' organ as initially thought (Lye and Chtarbanova, 2018; Contreras and Klämbt, 2023; Feng et al., 2024). In contrast to the overexpression of AMPs, another important immune-related gene, *lysozyme* (LOC724899), was downregulated in bees carrying high viral loads from the Colony A dataset, showing a strong inverse relationship with viral load in our correlation analyses. Lysozyme's

role as an antibacterial, anti-viral, and immunomodulatory agent is well established in both vertebrates and invertebrates (Ragland and Criss, 2017; Chen et al., 2018; Liu et al., 2023; Feng et al., 2024), and our findings align with previous observations that DWV infection can suppress lysozyme-mediated immunity in honey bees (Yang and Cox-Foster, 2005; Ray et al., 2025), likely by interfering with NF-κB signalling (Di Prisco et al., 2016). This dampening of the immune response mirrors similar observations reported for both invertebrates (Shelby et al., 1998; Palmer et al., 2019) and vertebrates (Pang et al., 2000), where infections by viruses have been linked to suppression of immune functions. The inverse relationship between viral load and *lysozyme* expression suggests a tightly linked mechanism by which DWV weakens the host's innate defences. In addition, we detected another set of genes generally well known for their role as immune regulators in insects, that were differentially expressed in the mushroom bodies of bees in our study: LOC100578156, orthologue of *Drosophila pirk*, a negative regulator of the immune deficiency pathway (Kleino et al., 2008), was upregulated in the Colony A group. Flies that overexpress *pirk* are more susceptible to gram-negative bacterial infection, and the same gene has previously been identified as an immune effector in *A. mellifera*, in which it is overexpressed in bees with Black queen cell virus (Doublet et al., 2016). *Phenoloxidase subunit A3* (*PPO*) was significantly downregulated in Colony A bees with high viral loads. *PPO* is a well characterised and crucial component of the insect immune system (González-Santoyo and Córdoba-Aguilar, 2012). Its expression is known to vary according to developmental stage, where it is typically upregulated in white-eyed pupae, downregulated in brown-eyed pupae, and shows varying regulation in adults (Kuster et al., 2014; Khongphinitbunjong et al., 2015; Zaobidna et al., 2015; Tesovnik et al., 2019; Pizzorno et al., 2021). Intriguingly, LOC100577198, an uncharacterised protein-coding gene that was downregulated in the Poor Learner group in the present study, has also previously been identified as an AMP involved in detoxification and immunity (Wu et al., 2023).

In the Poor Learners dataset, we noted several downregulated genes associated with neural functions in insects and other invertebrates: *Crustacean cardioactive peptide receptor*, a highly conserved insect neuropeptide responsible for varied biological functions in a range of species (Stangier et al., 1987; Zhou and Nagata, 2021; Gilbert et al., 2025;); *Homeotic Protein Deformed*, a Hox transcription factor that is crucial during embryonic development (Abzhanov and Kaufman, 1999; Feng et al., 2021), which has been observed upregulated in response to insecticide exposure (Fent et al., 2020b); *Krüppel-like factor 10*, a transcriptional regulator which has been implicated in immune modulation (Kaczynski et al., 2003; Kulkarni et al., 2022); and *Hbg3*, which encodes the alpha-glucosidase enzyme, a component of carbohydrate metabolism with expression associated with the transition from nursing to foraging. Interestingly, the upregulation of *Hbg3* has also been noted in precocious foragers (Ueno et al., 2015; Colin et al., 2019; Traniello et al., 2020), and while other studies have shown both its upregulation (Dussaubat et al., 2012; Fent et al., 2020a) and downregulation (Chen et al., 2021b; de Castro Lippi et al., 2025) in response to insecticides and pathogens, its downregulation here in affected bees may demonstrate how viral infection can impact foraging.

Although the present study did not seek to identify lncRNAs, several were noted in both the Poor Learner and Colony A datasets. The expression of two of these lncRNAs showed strong correlation with viral load, further demonstrating their importance in response to DWV infection. It is well established that lncRNAs are associated with a range of biological functions, including regulation (Wang and Chang, 2011), and the results of our GRN analysis are

intriguing, particularly the presence of *apidaecin* and *argonaute-2* as potential targets of LOC113218549. *Argonaute-2* is a key component of the short-interfering RNA (siRNA) pathway, involved in antiviral defence in invertebrates (Brutscher and Flenniken, 2015), and its upregulation has been observed in response to DWV infection (Brutscher et al., 2017; Zhao et al., 2019; Pizzorno et al., 2021; Norton et al., 2025) and other viruses (Brutscher and Flenniken, 2015; Galbraith et al., 2015). The observation that in both datasets the lncRNA LOC113218549 is the most strongly significantly differentially expressed gene after *apidaecin* is compelling.

Some limitations of this study should be recognised. First, our analysis was limited to two colonies, and as such caution should be exercised when generalising our findings to colonies existing in a variety of genetic and ecological contexts. We noted a distinct effect of colony of origin on differential gene expression: this clear separation between Colony A and Colony B samples is likely due to a combination of varying factors, such as quality of forage, *Varroa* presence/abundance and DWV loads, and the unique genetic background that characterises each honeybee colony – due to the different origin of the queen and the males she mated with. Additionally, Colony B received miticide treatments, which may have affected the mode of transmission in the colonies – by *Varroa* mites in Colony A and oral transmission in Colony B. Future studies could incorporate a wider range of settings in order to further clarify the impact of these factors. Second, we examined the bees' responses to a relatively simple associative learning task. More-refined or -advanced challenges may reveal subtle or nuanced impacts of DWV on cognition that this study has failed to uncover. Third, we have considered a single time point post-infection, which cannot capture the probable varying temporal transcriptomic responses to DWV infection: these responses might not only vary progressively over time, as a result of changing physiological conditions, but also in a cyclical fashion, as a consequence of natural circadian rhythms that are known to play a key role in the overall regulation of gene expression in insects (Sandrelli et al., 2008). Of note, we could not control for the age of the foragers that we sampled, another factor that has the potential to influence the transcriptomic response to a viral infection. Tagging individual bees and testing cohorts of foragers of comparable age is challenging but could surely be attempted in the future. Finally, our discovery of lncRNAs is intriguing, but further work will be required in order to validate the functional roles of these in the honey bee brain.

### Conclusions
The varying results obtained in this study – partially unexpected – demonstrate the importance of considering a combination of pathogen load, transmission route, treatments, neurotranscriptomic responses, and behaviour in order to understand the impact of viral pathogens on cognition. Our work highlights the benefits of analyses carried out on bees with naturally occurring DWV infections over the artificial induction of infections either with abdominal injections or by feeding. A comparison of the effects of these two routes of infection, injection versus feeding, in a more natural scenario would be informative, since it would facilitate future studies that wish to examine the effects of natural infections. Moreover, natural infections that present extremely high DWV loads – such as those observed in one of the two groups in our study – normally start during the larval or pupal stages, often mediated by the action of *Varroa* mites feeding on these individuals, and therefore have the potential to affect the developing brain. This is unlikely to happen in studies involving artificial infection via DWV injection or feeding to adult bees, the

brains of which are already fully developed at the moment of treatment. We have also emphasised how our approach here has further refined studies of infection and gene expression in the honey bee brain, by focussing on the mushroom bodies. A further refinement would be to use single-cell sequencing to compare infected and non-infected neurons within the mushroom bodies.

## MATERIALS AND METHODS

### Honey bee samples

The honey bee samples used in this study were obtained from two colonies located in two apiaries near Aberdeen, Scotland, 20 km apart from each other: Cruickshank Botanic Garden, on the University of Aberdeen Kings College campus (Grid Reference NJ936085), and Newburgh (NJ998260). These colonies derive from an initial set of commercially acquired queens – used to establish the two apiaries – that were a cross of *Apis mellifera carnica* and *Apis mellifera mellifera*. Colonies in Newburgh (Colony A) typically receive minimal *Varroa* treatments – just once a year – and therefore suffer from relatively high *Varroa* and DWV levels (Woodford et al., 2022). In contrast, colonies in Cruickshank Botanical Garden (Colony B) receive standard treatments against the parasitic mite *Varroa destructor*, and consequently have generally lower *Varroa* infection rates and lower DWV levels (Fig. 1A). We intentionally targeted colonies located in these two apiaries to obtain honey bee samples with expected significant variation of natural viral loads. Honey bee foragers were collected in the summer of 2020 between 09:00 am and 10:00 am at the colony entrance when returning from a foraging trip. The bees were transported individually in Petri dishes, and once in the laboratory were immobilised on ice and harnessed for an absolute conditioning assay via PER (Fig. 1B). Harnesses were made from perspex tubing (approximately 8 mm in diameter and 40 mm in height), and a 15 mm incision was made on the side of the tube. Individual bees were placed in the harness where the incision was made – wings facing outwards – and a thin strip of adhesive tape was secured around the bee's neck ensuring the bee was still able to freely move mouthparts and antennae.

### PER conditioning assays

This set of experiments was performed to quantify the learning performance of individual bees and thereby allocate each to one of two distinct behavioural groups: Good Learners and Poor Learners. Harnessed bees were initially fed with 3–5 µl of 30% (w/w) sucrose solution and desensitised to water by touching the antennae with a soaked toothpick. They were then held in the harnesses for one hour before the start of the assays, in the dark (within a lab cabinet) and at room temperature (between 22 and 25°C), with humidity ranging between 35% and 40%. These conditions were also used to house bees overnight (see below). The general PER conditioning protocol followed the approach of Mota and Giurfa (2010), but with only one scent being used and no reversal performed. The bees were conditioned to respond to citral (Sigma-Aldrich, St Louis, MO, USA) through a five-time exposure to the odour with reinforcement by touching the antennae with a 30% (w/w) sucrose solution. 5 µl citral (95%, v/v) was pipetted onto a piece of filter paper, which was then inserted into a 20 ml syringe. The bees were exposed to citral by expressing the full volume of air from the syringe around 0.5 cm away from the antennae. Foragers that exhibited PER to the first citral exposure were discarded, as these bees were responding randomly to the stimulus at this point, since the conditioning phase had not yet taken place. If no PER was recorded for the odour itself during the first trial, the stimulus was reinforced with a 30% (w/w) sucrose solution, the reward. Individuals not responding to sucrose were discarded before further testing, as it would have been impossible to proceed with the conditioning phase with these individuals.

After conditioning, but still on the same day of capture, bees were re-fed and exposed to the same odour three times without reinforcement – this was repeated the following day for a total of six tests. Samples were then allocated to the two behavioural groups according to PER response in the non-reinforced trials as follows: Poor Learners=no response in any of the trials; Good Learners=PER response in all six trials. Thereafter, bees were frozen in a −80°C freezer and stored there until later processing for molecular work, which occurred a few months after the end of the field season.

**Table 1. Summary of the 33 mushroom body samples used in this study**

| Sample ID | Viral load (log$_{10}$) | Viral load* | Performance‡ | Colony§ |
|---|---|---|---|---|
| 1 | 2.95 | low | poor | B |
| 2 | 2.51 | low | poor | B |
| 3 | 2.59 | low | poor | A |
| 4 | 2.62 | low | poor | A |
| 5 | 3.96 | low | good | A |
| 6 | 2.49 | low | good | A |
| 7 | 4.01 | low | good | B |
| 9 | 3.56 | low | poor | B |
| 10 | 5.18 | high | poor | B |
| 11 | 2.68 | low | poor | A |
| 12 | 10.35 | high | poor | A |
| 13 | 4.05 | low | good | B |
| 14 | 9.83 | high | good | B |
| 15 | 2.75 | low | good | A |
| 16 | 9.25 | high | good | A |
| 17 | 2.49 | low | poor | B |
| 18 | 2.81 | low | poor | A |
| 19 | 3.02 | low | good | A |
| 20 | 2.38 | low | good | B |
| 21 | 9.36 | high | poor | A |
| 22 | 6.71 | high | poor | B |
| 23 | 6.47 | high | good | A |
| 24 | 9.02 | high | good | B |
| 25 | 2.52 | low | poor | B |
| 26 | 2.88 | low | poor | A |
| 27 | 2.98 | low | good | A |
| 28 | 2.44 | low | good | B |
| 29 | 10.35 | high | poor | A |
| 30 | 5.14 | high | poor | B |
| 31 | 5.28 | high | good | A |
| 32 | 6.38 | high | good | B |
| 33 | 5.26 | high | good | B |
| 34 | 5.35 | high | good | B |

Note that sample #8 is missing due to insufficient yields obtained after RNA isolation.
*High=DWV GE ≥log$_{10}$ 5; low=DWV GE ≤log$_{10}$ 4 (total quantity of virus in the mushroom bodies). ‡Poor=Poor Learners that failed all trials; good=Good Learners that passed all trials. §Colony A=bees collected from the apiary in Newburgh; Colony B=bees collected from the apiary in Cruickshank.

### Processing of samples for molecular work

Total RNA was isolated from the mushroom bodies of individual bees for two purposes: (1) to quantify DWV loads in this tissue and (2) to perform transcriptomic profiling with RNA-seq. Mushroom bodies were dissected on dry ice under a stereomicroscope following (Szymański et al., 2024), and homogenised in a Tissue Lyser II device (Qiagen, Hilden, Germany) using 1 ml TRIzol and 2.3 mm zirconia beads (Thistle Scientific, Glasgow, UK). We then used a standard TRIzol protocol – following the manufacturer's instructions – to obtain total RNA that was resuspended in 20 µl of nuclease-free water. These RNA samples were used for the quantification of DWV

**Table 2. The six mushroom body samples with reads including overrepresented sequences**

| Sample ID | Viral load* | Before (%) | After (%) | Source |
|---|---|---|---|---|
| 12 | High | 9.1 | 0 | |
| 14 | High | 11.1 | 0 | |
| 16 | High | 1.0 | 0 | |
| 20 | Low | 0.2 | 0 | |
| 21 | High | 0.6 | 0.6 | Ribosomal RNA |
| 29 | High | 2.2 | 0 | |

Reads aligning to the DWV genome were removed, and the percentage of overrepresented sequence before and after removal is shown. The expected source of any remaining overrepresented sequences (according to a BLAST search) is also noted. *High=DWV GE ≥log$_{10}$ 5; low=DWV GE ≤log$_{10}$ 4.

**Table 3. List of genes differentially expressed in mushroom bodies of Poor Learners when comparing individuals with high and low viral loads (DWV≥log$_{10}$ 5 and ≤log$_{10}$ 4, respectively). The same analysis revealed no difference in gene expression for Good Learners with high versus low DWV loads (all P-values>0.05)**

| Regulation* | OGSv3.2 ID‡ | OGSv1.x ID§ | Gene ID¶ | Gene description | Log$_2$ FC** | P‡‡ | Rho§§ |
|---|---|---|---|---|---|---|---|
| Up | GB46236, GB47546, GB51306 | GB13473 GB17354, GB17782 | 406140, *Apid1* | apidaecin 1 | 5.69 | $5.63×10^{-6}$ | 0.34 |
| | | | LOC113218549 | non-coding RNA | 4.74 | $5.63×10^{-6}$ | 0.58 |
| | GB1223 | GB17538 | LOC406142 | hymenoptaecin | 3.35 | $2.22×10^{-2}$ | 0.27 |
| Down | GB48020 | GB13606 | LOC102655185 | serine-rich adhesin for platelets-like | 3.57 | $3.93×10^{-2}$ | 0.65 |
| | GB42311 | GB14425 | LOC102654257 | sosie oogenesis-related protein sosie | 2.99 | $2.31×10^{-2}$ | 0.65 |
| | | | LOC107965291 | non-coding RNA | 2.81 | $2.22×10^{-2}$ | 0.75 |
| | GB43247 | GB19017 | LOC406131, *Hbg3* | alpha-glucosidase | 2.36 | $3.58×10^{-2}$ | 0.61 |
| | GB51299 | GB13409 | LOC724252, *Dfd* | homeotic protein deformed | 2.06 | $2.32×10^{-2}$ | 0.69 |
| | GB41033, GB41034 | GB11097, GB14306 | LOC726505 | uncharacterised protein coding | 1.58 | $3.34×10^{-2}$ | 0.77 |
| | GB45040 | GB10114 | LOC410326 | Krüppel-like factor 10 | 1.43 | $3.11×10^{-3}$ | 0.61 |
| | GB40771 | GB13235 | LOC412829 | arylsulfatase B | 1.42 | $2.92×10^{-2}$ | 0.55 |
| | GB54316 | GB19289, GB30535 | LOC726935 | Crustacean cardioactive peptide receptor | 1.32 | $3.27×10^{-2}$ | 0.55 |
| | GB52955 | GB11664 | LOC411159 | uncharacterised protein coding | 1.23 | $3.11×10^{-3}$ | 0.76 |
| | GB49376 | GB19733 | LOC724642 | F-box/LRR-repeat protein 3 | 1.20 | $2.71×10^{-2}$ | 0.79 |
| | GB55973 | | LOC100577198 | uncharacterised protein coding | 1.05 | $2.22×10^{-2}$ | 0.39 |

*Up=more highly expressed in Poor Learners with high viral loads; Down=lower expression in Poor Learners with high viral loads. ‡Gene IDs as specified in Official Gene Set (OGS) version 3.2 (Elsik et al., 2014). §Gene IDs as specified in OGS version 1.x (Honeybee Genome Sequencing Consortium, 2006). ¶NCBI Gene ID (https://www.ncbi.nlm.nih.gov/gene/). **Log$_2$ FC=absolute difference in gene expression expressed as fold changes on a log$_2$ scale (|log$_2$FC|). ‡‡Bonferroni-adjusted P-value. §§Rho=correlation coefficient between normalised read counts and DWV loads.

loads with an RT-qPCR assay: 300 ng of RNA was converted into cDNA using the iScriptTM cDNA Synthesis Kit (Bio-Rad Laboratories Inc., Hercules, CA, USA) and following the manufacturer's protocol. cDNA samples (300 ng) were used as templates to run RT-qPCR reactions with primers capable of detecting the two most common strains of DWV (Bradford et al., 2017). DWV loads were quantified as genome equivalents (GE) using calibration curves obtained from known quantities of DWV GE inserted into plasmids.

Bees were allocated to two groups as follows: high DWV – when mushroom bodies harboured more than log$_{10}$ 5 DWV GE overall (accuracy to one decimal point), and low DWV – when viral loads were lower than or equal to log$_{10}$ 4 DWV GE overall. By combining the behavioural grouping with the grouping resulting from DWV analyses, we were able to obtain a total of 33 bee mushroom body samples (Table 1) that underwent RNA-seq profiling. These samples were processed via a clean-up and concentration step that also included the removal of residual genomic DNA (Zymo Research Corp., Irvine, CA, USA) and thereafter they were sent to the Centre for Genome Enabled Biology and Medicine at the University of Aberdeen for library preparation and sequencing.

Library prep was performed using a TruSeq Stranded mRNA-seq Kit: the 33 samples were split into two batches (balanced by treatment groups and colony of origin) and indexed for Illumina sequencing. Sequencing was carried out on a NextSeq500 v2.5 platform using Illumina High Output v2.5 75 cycle flowcell and reagents. The output produced a total of 874.2 M single-end 75 bp reads – on average 23.2 M per sample.

### Processing of RNA-seq data

Raw reads were trimmed using Trim Galore! v0.6.4 (Krueger, 2019). Illumina adapter sequences were removed using a stringency setting of 3; ends including bases with a Phred score less than 30 were removed; and reads shorter than 30 base pairs were discarded. Read quality was then assessed with FastQC v0.11.8 (Andrews, 2010).

There was no evidence of adaptor contamination, but six samples – five of which carried high viral loads – included overrepresented sequences, lower-than-average alignments, and slightly atypical per-sequence GC content plots, suggesting contamination. We extracted these overrepresented sequences and queried them against the NCBI nucleotide database using BLAST (Altschul et al., 1990), which revealed close matches to DWV genomes.

**Table 4. List of genes differentially expressed in bees from Colony A when comparing individuals with high versus low viral loads (DWV≥log$_{10}$ 5 and ≤log$_{10}$ 4, respectively)**

| Regulation* | OGSv3.2 ID‡ | OGSv1.x ID§ | Gene ID¶ | Gene description | Log2 FC** | P‡‡ | Rho§§ |
|---|---|---|---|---|---|---|---|
| Up | GB46236, GB47546, GB51306 | GB13473, GB17354, GB17782 | 406140, *Apid1* | apidaecin 1 | 5.81 | $8.16×10^{-8}$ | 0.63 |
| | | | LOC113218549 | long non-coding RNA | 5.70 | $6.13×10^{-12}$ | 0.68 |
| | GB1223 | GB17538 | LOC406142 | hymenoptaecin | 3.79 | $2.03×10^{-4}$ | 0.53 |
| | GB47318 | GB18323 | LOC406144 | abaecin | 2.95 | $2.72×10^{-2}$ | 0.55 |
| | | | LOC113218947 | long non-coding RNA | 2.31 | $1.28×10^{-3}$ | 0.74 |
| | | | LOC102654076 | long non-coding RNA | 1.35 | $2.00×10^{-2}$ | 0.59 |
| | GB44455 | GB15528 | LOC100578156 | uncharacterised protein coding | 1.32 | $4.07×10^{-6}$ | 0.45 |
| | GB41972 | GB13568 | LOC551263 | monocarboxylate transporter 13 | 1.17 | $3.64×10^{-3}$ | 0.50 |
| Down | GB40164 | GB15106 | LOC724899 | lysozyme | 1.84 | $2.12×10^{-7}$ | 0.83 |
| | GB43738 | GB18313 | LOC406155, *PPO* | phenoloxidase subunit A3 | 1.77 | $1.28×10^{-3}$ | 0.66 |
| | | | LOC102655375 | long non-coding RNA | 1.35 | $9.03×10^{-3}$ | 0.64 |
| | GB42891 | GB16731 | LOC102655319 | inner centromere protein A | 1.34 | $3.14×10^{-2}$ | 0.55 |
| | GB49476 | GB13360 | LOC408544 | lysyl oxidase homolog 4 | 1.09 | $2.34×10^{-3}$ | 0.75 |

*Up=more highly expressed in Colony A bees with high viral loads; Down=lower expression in Colony A bees with high viral loads. ‡Gene IDs as specified in OGS version 3.2 (Elsik et al., 2014). §Gene IDs as specified in OGS version 1.x (Honeybee Genome Sequencing Consortium, 2006). ¶NCBI Gene ID (https://www.ncbi.nlm.nih.gov/gene/). **Log$_2$ FC=absolute difference in gene expression expressed as fold changes on a log$_2$ scale. ‡‡Bonferroni-adjusted P-value. §§Rho=correlation coefficient between normalised read counts and DWV load.

**Table 5. Genes identified as potential targets of lncRNA LOC113218549**

| Group | OGSv3.2 ID* | OGSv1.x ID‡ | Gene ID§ | Gene description |
|---|---|---|---|---|
| Both | GB46236, GB47546, GB51306 | GB13473, GB17354, GB17782 | 406140, *Apid1* | apidaecin 1 |
| Poor Learners | GB50955 | GB15464 | 411577, *Ago2* | protein argonaute-2 |
| | GB41972 | GB13568 | 551263 | monocarboxylate transporter 13 |
| | GB55508 | GB13402 | 408656 | RNA exonuclease 1 homolog |
| | GB43565 | GB10235 | 724338 | OTU domain-containing protein 5-B |
| | GB52993 | GB14643 | 724973 | focal adhesion kinase 1, |
| | GB45597 | GB18319 | 412679 | rap1 GTPase-GDP dissociation stimulator 1-B |
| | GB49868 | GB19653 | 100578674 | disks large-associated protein 5 |
| | GB44455 | GB15528 | 100578156 | uncharacterised protein coding |
| | | | 100576603 | uncharacterised protein coding |
| | | | 107965519 | long non-coding RNA |
| Colony A | GB41283 | GB10310 | 408864 | waprin-Phi1 |

*Gene IDs as specified in OGS version 3.2 (Elsik et al., 2014). ‡Gene IDs as specified in OGS version 1.x (Honeybee Genome Sequencing Consortium, 2006).
§NCBI Gene ID and name (if available) (https://www.ncbi.nlm.nih.gov/gene/).

This prompted us to investigate whether DWV sequences were also present in the other RNA samples – as well as sequences associated with other seven commonly occurring honey bee RNA viruses (McMenamin and Flenniken, 2018). We used HISAT2 v2.1.0 (Kim et al., 2015) to align reads from each sample to two DWV reference genomes – a recombinant strain (NCBI accession MN538210) (Norton et al., 2020) and a DWV-B strain (NCBI accession MN565038) (Dalmon et al., 2019).

DWV RNA was present in many samples, but was several orders of magnitude greater in the six samples mentioned above and in one other sample that did not suffer overrepresented sequences (Table S4). We also aligned all samples to the reference genomes of Chronic Bee Paralysis Virus, Sacbrood Virus, Black Queen Cell Virus, Israeli Acute Paralysis Virus, Kashmir Bee Virus, Acute Bee Paralysis Virus, and Slow Bee Paralysis Virus. Alignment rates revealed the absence of these viruses in our samples, supporting observations from previous years in the same apiaries. We proceeded therefore with the alignment of the sequencing output to the *A. mellifera* genome (assembly Amel_HAv3.1; NCBI Assembly: GCF_003254395.2), but for those six samples with overrepresented sequences we removed all DWV-aligned reads – ensuring consistency across the set (Table 2), and gene-level read counts were obtained using FeatureCounts (from Subread v1.6.2) (Liao et al., 2014) for downstream analyses.

## Gene expression analyses

We performed exploratory analyses on the RNA-seq dataset with PCA to investigate variations in patterns of gene expression due to colony location, learning performance, and viral load. In order to reduce noise, enhance biological signal, and to improve visualisation and clustering (Love et al., 2014; Satija et al., 2015; Soneson and Robinson, 2018), analyses were conducted on the normalised gene expression counts for the 500 most-variable of the 12,332 genes from the 33 mushroom body samples. Raw counts were normalised using the median of ratios method and applying a variance stabilising transformation (Anders and Huber, 2010), and a Pearson correlation coefficient matrix was created for the pairwise comparison of samples. Noting greater clustering amongst Colony A samples, we quantified clustering strength by measuring silhouette width (Rousseeuw, 1987) and the within-colony Euclidean distances in PCA space.

We carried out differential gene expression analysis using DESeq2 v1.40.2 (Love et al., 2014). To ensure that only genes with sufficiently large read counts across samples were used in the analysis, we filtered the list of genes according to expression level by using the filterByExpr function from edgeR v 3.42.4 (Robinson et al., 2010), and then used DESeq2 to reveal a list of differentially expressed genes with a Benjamini-Hochberg-adjusted $P$-value<0.05 and an absolute $\log_2$ fold change in expression of one or more ($|\log_2$FC|>1). For each differentially expressed gene, we also examined the Spearman's correlation coefficient between absolute viral load (see Table 1) and the normalised gene expression level. We also used Bioconductor's EnrichmentBrowser v 2.30.2 (Geistlinger et al., 2016) to assemble gene sets from the expression data using overrepresentation analysis (Goeman and Bühlmann, 2007) and tested for enrichment of functional gene sets as defined in GO (Ashburner et al., 2000) and KEGG (Kanehisa et al., 2008)

pathway annotations. Finally, we used BioNero's (Almeida-Silva and Venancio, 2022) exp2grn function to explore GRNs, using differentially expressed lncRNAs as potential regulators of all genes.

## Acknowledgements

We would like to thank Ewan Campbell for beekeeping support and the Centre for Genome Enabled Biology and Medicine at the University of Aberdeen for sequencing and processing of raw data. We would also like to thank the two anonymous reviewers and the editor for providing comments that helped improve the quality and clarity of the manuscript.

## Competing interests
The authors declare no competing or financial interests.

## Author contributions
Conceptualization: S.E.L., A.S.B., F.M.; Data curation: S.E.L.; Formal analysis: S.E.L.; Funding acquisition: A.S.B., F.M.; Investigation: S.E.L., L.D.; Methodology: S.E.L., L.D.; Project administration: L.D., A.S.B., F.M.; Resources: A.S.B., F.M.; Software: S.E.L.; Supervision: A.S.B., F.M.; Validation: S.E.L., L.D.; Visualization: S.E.L.; Writing – original draft: S.E.L., F.M.; Writing – review & editing: S.E.L., L.D., A.S.B., F.M.

## Funding
This research was funded by an Internal Funding to Pump-Prime Interdisciplinary Research and Impact Activities, awarded to F.M. and A.S.B. by the Aberdeen Grant Academy, and it was also supported by the CB Dennis British Beekeepers' Research Trust, as part of a grant awarded to F.M. and A.S.B. Open Access funding provided by University of Aberdeen. Deposited in PMC for immediate release.

## Data and resource availability
The original data presented in the study are available in the NCBI Sequence Read Archive (SRA) at https://www.ncbi.nlm.nih.gov/sra/PRJNA1261397. The code used for analysis is available via GitHub at https://github.com/simonedwardloughran/neurotranscriptomics-dwv. All other relevant data and details of resources can be found within the article and its supplementary information.

## Peer review history
The peer review history is available online at https://journals.biologists.com/bio/lookup/doi/10.1242/bio.062204.reviewer-comments.pdf

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
