## [Peer Review File · Biology Open]

Neurotranscriptomic profiling of Deformed Wing Virus-infected honey bee foragers (*Apis mellifera*) with different cognitive abilities

Simon E. Loughran, Lauren Dingle, Alan S. Bowman and Fabio Manfredini

DOI: 10.1242/bio.062204

Editor: Lewis Halsey

Review timeline

Original submission:	8 August 2025
Editorial decision:	18 August 2025
First revision received:	20 October 2025
Accepted:	21 October 2025

Original submission

First decision letter

MS ID#: bio.062204

MS Title: Neurotranscriptomic profiling of Deformed Wing Virus-infected honey bee foragers (*Apis mellifera*) with different cognitive abilities

Authors: Simon E. Loughran; Lauren Dingle; Alan S. Bowman; Fabio Manfredini

Article Type: Research Article

I have now reached a decision on the above manuscript.

The reviewer reports are shown at the bottom of this email or can be accessed, together with a copy of this decision letter, by going to:

As you will see, the reviewers raised a number of substantial criticisms that prevent me from accepting the paper at this stage. Moreover, I have concerns over the power of your study and the claims made of 'no effect' based on high p values. It is wrong to state 'there is no correlation' or 'there is no difference' when the p value is high. Fisher himself made clear that a p value does not provide evidence for the null hypothesis, regardless of how 'non-significant' it is; in such cases we can only conclude that we find our data unsurprising if we believe in the null hypothesis. Because the p value assumes the null hypothesis is true it cannot therefore evaluate the likelihood that this is so. What to do?

- 1) Report exact p values not e.g. $p > 0.05$
- 2) Don't interpret $p > 0.05$ as no effect.
- 3) If you really want to make conclusions based on no (meaningful) effect then you will need to undertake e.g. equivalence tests, or alternatively e.g. calculate Bayes factors. Daniel Laken's online stats book is helpful here, e.g. 9 Equivalence Testing and Interval Hypotheses - Improving Your Statistical Inferences

The reviewers and I suggest, however, that a revised version might prove acceptable, if you can address the concerns. If you think that you can deal satisfactorily with the criticisms on revision, I would be pleased to see a revised manuscript. We would then return it to the reviewers.

At this stage, we also ask you to ensure your manuscript complies with our formatting guidelines. Provided you are able to fully address the referees' comments, we are positive about publication of your paper (we accept over 95% of revision submissions) and therefore hope you won't mind any extra work involved in reformatting your manuscript at this point.

Please ensure that you clearly highlight all changes made in the revised manuscript. Please avoid using 'Tracked changes' in Word files as these are lost in PDF conversion.

I should be grateful if you would also provide a point-by-point response detailing how you have dealt with the points raised by the reviewers in the 'Response to Reviewers' box. Please attend to all of the reviewers' comments. If you do not agree with any of their criticisms or suggestions please explain clearly why this is so.

Reviewer 1

Loughran et al., investigates the transcriptomic responses of *Apis mellifera* mushroom bodies during naturally occurring infections with Deformed Wing Virus (DWV), focusing on the relationship between molecular changes and cognitive performance in an associative learning task. By specifically targeting mushroom bodies, which are implicated in high-order neural processing in insect, the authors employ a spatially refined transcriptomic approach that sheds light on host-pathogen interactions in neural tissue.

Key findings of the study include differential expression of immune-related genes, alteration of antimicrobial peptides, changes in stress-related pathway, and novel insights into the potential regulatory role of long non-coding RNAs. Importantly, the authors report no correlation between DWV load in mushroom bodies and cognitive performance, challenging conclusion from earlier studies involving artificially induced infections.

The study is interesting and relevant to the journal's readership, and the data generally support the conclusion. However, improvements in data presentation, methodological clarity, and manuscript organization would substantially strengthen the work. Suggestions below:

- a. The rationale for using *Apis mellifera* from geographically and genetically different background is unclear. Given the potential variability introduced by these differences, please clarify why this approach was taken and discuss implications for interpreting and generalizing the findings.
- b. The authors narrowed down the analysis to Colony A, but it is unclear whether the sample sizes for behavior (poor and good learners) and DWV load (low and high) provide adequate statistical power. Also note that forager bees were collected within a one-hour window, factors such as age, foraging experience could still vary and impact both behaviors and viral load.
- c. Including a clear schematic of the experimental design would help readers.
- d. Provide high-resolution statistical figures with accessible and reader-friendly for better data visualization. In the differential gene expression analysis, clearly label comparison groups and ensure that axis labels are legible at publication size.
- e. State a brief rationale at the beginning of each results subsection to frame the purpose of the analysis being presented. The discussion is too lengthy, contains repeated descriptions of results, and would benefit from moving some interpretive content to the result sections.

Minor comments:

1. Replace honeybee in the title with *Apis mellifera* and expand DWV.

2. Italicize genus and species names throughout the manuscript.
3. Expand all abbreviations at first use and ensure consistency thereafter.
4. In keywords, consider using transcriptomics rather than neurotranscriptomics.
5. Revise for overly long or complex sentences, and correct typographical errors - for example, remove log (line 954), bee (line 971), brackets for the reference (lines 181- 182), etc.
6. Provide references for statement in lines 101-104.
7. Line 164, include units. AM?
8. Line 179, clarify how assays were performed in darkness.
9. Specify the recommended room temperature for assays.
10. Provide catalog or product number for all reagents and instruments used in this study.
11. Line 184, clearly state the concentration of citral odor used.
12. Lines 191-193, assay steps are confusing, needs clarification.
13. Line 196, what is the author recommended time duration for freezing?
14. Sample ID 8 is missing- clarify whether it was intentionally excluded or if this is an error in numbering.
15. Report the sample numbers for each group in the figure legends.

Reviewer 2

This manuscript investigates two colonies of honeybee foragers looking for links among (a) natural levels of viral infections, (b) transcriptomic variations, and (c) cognitive abilities. Towards this goal, they tested animals in an associative learning assay and then harvested their mushroom bodies to determine viral load and transcriptomic profile.

Curiously, the authors find little or no evidence for associations between cognitive ability vs. viral load or cognitive ability vs. changes in gene expression. This negative result runs counter to several prior studies done in laboratory honeybee strains. In contrast, their studies uncover several genes and RNA that correlate (negatively or positively) with viral load.

Several questions arise:

(1) In data where learning ability is a factor, only Poor learners are discussed (e.g. Table 3). Why are data from Good learners overlooked in these comparisons? Is it possible that correlations seen in Poor learners are reversed in Good learners? As the authors note, DWV has been shown to paradoxically improve and impair cognitive function in honeybees. As Good learners are not included in the main Tables, I am wondering if they have nothing to offer in this study? Along these lines, I am unsure if having "...cognitive abilities" in the manuscript is meaningful.

(2) The animals were collected during 9-10 am. Could the authors comment on the choice of the collection time? I am wondering how the time-of-day could be affecting cognitive ability, DWV load and transcription profiles, given the circadian clock controls much of honeybee physiology.

(3) Since most of the paper focuses on Colony A, it would be helpful to have PCA analysis done on just Colony A data. The analysis could be drastically different because without Colony B, the data have much less variance. I wonder how the single colony data would cluster.

(4) Since the natural DWV load has no clear effect on learning as assessed through PER, could the authors speculate on what type of effect on cognition might these natural levels of virus have in these bees? For example, if an assay measured collective cognition would Colony A fare worse than Colony B insects?

Minor comments:

(1) Figures 2 and 4 missing x-axis label.

(2) Figure 5: Would be clearer if panel titles such as "(A) F-box/LRR..." appeared above each scatter plot instead of below.

(3) Line 422: "Both the Colony A and Poor..." What do the authors mean by scale-free topology and why is that significant?

(4) The current Figure 1 shows that gene expression data cluster in terms of colony of origin and Figure 3 shows that Colony A gene expression has more uniformity. To me, these two data belong together, and might be helpful if they appeared in the same figure or one right after the other (if in separate figures). Right after presenting these data, the authors would focus on Colony A data for the rest of the paper. Why not get these out of the way in the beginning?

(5) Figure S3: For each panel, please enlarge font size for the gene ID and corresponding stat metrics. Currently, they are hard to read.

Reviewer's Responses to Questions

Experimental quality

Does each figure have the proper controls?

If 'No', please indicate reasons in Comments for Author box below.

Reviewer #1:

- Yes

Reviewer #2:

- Yes

Were the data analyzed using appropriate statistical tests?

If 'No', please indicate reasons in Comments for Author box below.

Reviewer #1:

- Yes

Reviewer #2:

- Yes

Reproducibility

Were experiments performed using adequate number of biological replicates?

If 'No', please indicate reasons in Comments for Author box below.

Reviewer #1:

- No

Reviewer #2:

- Yes

Does the methods section provide sufficient detail to permit reproducibility?

If 'No', please indicate reasons in Comments for Author box below.

Reviewer #1:

- Yes

Reviewer #2:

- Yes

Completeness

Are the manuscript's conclusions supported by the data?

If 'No', please indicate reasons in Comments for Author box below.

Reviewer #1:

- Yes

Reviewer #2:

- Yes

Scholarship

Do the authors cite and discuss the merits of data that would argue for and against their conclusion?

If 'No', please indicate reasons in Comments for Author box below.

Reviewer #1:

- Yes

Reviewer #2:

- Yes

Does the manuscript title & abstract accurately reflect the contents of the manuscript, without hyperbole?

If 'No', please indicate reasons in Comments for Author box below.

Reviewer #1:

- Yes

Reviewer #2:

- No

First revision

Author response to reviewers' comments

Dear Editor,

We were very pleased to see that our paper was well received by yourself and by the two anonymous reviewers, who provided insightful comments and suggestions that helped us improve the quality of the manuscript. In particular, we greatly appreciated comments like “The study is interesting and relevant to the journal's readership, and the data generally support the conclusion” (Reviewer 1) and the overall assessment of the quality of the research presented and the manuscript itself. We have addressed all points raised - including the point raised by yourself - in the text below, to the best of our knowledge, in blue. We have also reported the part of the text that was modified accordingly, where relevant, and we have highlighted in yellow the most significant modifications in the version of the manuscript that we are submitting together with this file. Please let us know if there are any further clarifications that you - or the reviewers - need.

Moreover, I have concerns over the power of your study and the claims made of 'no effect' based on high p values. It is wrong to state ‘there is no correlation’ or ‘there is no difference’ when the p value is high. Fisher himself made clear that a p value does not provide evidence for the null hypothesis, regardless of how ‘non-significant’ it is; in such cases we can only conclude that we find our data unsurprising if we believe in the null hypothesis. Because the p value assumes the null hypothesis is true it cannot therefore evaluate the likelihood that this is so. What to do?

- 1) Report exact p values not e.g. $p > 0.05$
- 2) Don't interpret $p > 0.05$ as no effect.
- 3) If you really want to make conclusions based on no (meaningful) effect then you will need to undertake e.g. equivalence tests, or alternatively e.g. calculate Bayes factors. Daniel Laken's online stats book is helpful here, e.g. 9 Equivalence Testing and Interval Hypotheses - Improving Your Statistical Inferences

Thank you for adding your personal comment and providing suggestions for how to improve the discussion of the outputs of our statistical analyses. With reference to the power of our study, we understand your concern but we believe that the sample size and the series of analyses associated with our study is appropriate to identify any major effect - if present. The initial set of 33 bees across 2 colonies is in line with previous studies in this area of research: both Pizzorno et al. (2021) and Dickey et al. (2023), for example, assessed 12 bees in total (a significantly smaller sample size than ours) while Li et al. (2019) assessed 6 pools of bee brains in total.

Dickey, Myra, Elizabeth M. Walsh, Tonya F. Shepherd, Raul F. Medina, Aaron Tarone, and Juliana Rangel. "Transcriptomic analysis of the honey bee (*Apis mellifera*) queen brain reveals that gene expression is affected by pesticide exposure during development." *Plos one* 18, no. 4 (2023): e0284929.

Pizzorno, Marie C., Kenneth Field, Amanda L. Kobokovich, Phillip L. Martin, Riju A. Gupta, Renata Mammone, David Rovnyak, and Elizabeth A. Capaldi. "Transcriptomic responses of the honey bee brain to infection with deformed wing virus." *Viruses* 13, no. 2 (2021): 287.

Li, Zhiguo, Tiantian Yu, Yanping Chen, Matthew Heerman, Jingfang He, Jingnan Huang, Hongyi Nie, and Songkun Su. "Brain transcriptome of honey bees (*Apis mellifera*) exhibiting impaired olfactory learning induced by a sublethal dose of imidacloprid." *Pesticide biochemistry and physiology* 156 (2019): 36-43.

This said, and because a similar point was also raised by Reviewer 1 (point b) we have added some considerations about sample size in the discussion, lines 479-483 “Perhaps the limited sample size (17 bees when considering Good Learners alone) could explain this outcome and a larger set of samples would increase the statistical power to detect significant differences; it must be noted though that the sample size we used is in line with previous studies on a similar topic (e.g. Pizzorno et al. 2021).”

As for your other points regarding the reporting of statistical analyses, we agree that giving too much weight on p-values is not suitable and that caution should be used when discussing results that do not support initial hypotheses. We have therefore followed your suggestion to report exact p-values (when possible) and to mitigate some of our conclusions regarding interpretation of results

with $p > 0.05$. See for example lines **366-368** “When comparing bees with high vs. low viral load within the subset of Good Learners (17 individuals in total) no genes were identified as significantly differentially expressed at our chosen threshold (Benjamini-Hochberg-adjusted $p = 0.05$).”

We would like to add though that in some instances we necessarily had to retain the use of the formula $p < 0.05$ or $p > 0.05$ when indicating a threshold or a range that allowed us to isolate groups of genes of interest. This is standard practice in the field of transcriptomics, as it provides a tool to focus on smaller groups of genes (that can be discussed further) according to how confident we are that their expression levels are biologically relevant and repeatable. This is extremely important when assessing many thousands of genes, as it would be impossible to give equal weight to all these elements - and of course we cannot report exact p -value for individual genes when there are so many.

Comments from the Reviewers:

Reviewer 1: Loughran et al., investigates the transcriptomic responses of *Apis mellifera* mushroom bodies during naturally occurring infections with Deformed Wing Virus (DWV), focusing on the relationship between molecular changes and cognitive performance in an associative learning task. By specifically targeting mushroom bodies, which are implicated in high-order neural processing in insect, the authors employ a spatially refined transcriptomic approach that sheds light on host-pathogen interactions in neural tissue.

Key findings of the study include differential expression of immune-related genes, alteration of antimicrobial peptides, changes in stress-related pathway, and novel insights into the potential regulatory role of long non-coding RNAs. Importantly, the authors report no correlation between DWV load in mushroom bodies and cognitive performance, challenging conclusion from earlier studies involving artificially induced infections.

The study is interesting and relevant to the journal's readership, and the data generally support the conclusion. However, improvements in data presentation, methodological clarity, and manuscript organization would substantially strengthen the work. Suggestions below:

Thank you very much for finding value in our work, for providing such a comprehensive overview of what we achieved and for deeming our manuscript suitable for the readership of the journal.

a. The rationale for using *Apis mellifera* from geographically and genetically different background is unclear. Given the potential variability introduced by these differences, please clarify why this approach was taken and discuss implications for interpreting and generalizing the findings.

Thank you for raising this point, which allows us to clarify what we mean by genetic background. In reality, both colonies used for this study derive from initial queens of a cross of *Apis mellifera carnica* and *Apis mellifera mellifera* that were initially used by us at the University of Aberdeen to establish colonies in both apiaries (Cruickshank and Newburgh). Therefore, the genetic background is not different in this sense. What we referred to in the paper is the genetic difference that naturally exists between individual honeybee colonies, being produced by different queens that mated with males of different origin (potentially). We clarified this in the the text:

lines 603-608 “We noted a distinct effect of colony of origin on differential gene expression: this clear separation between Colony A and Colony B samples is likely due to a combination of varying factors, such as quality of forage, Varroa presence/abundance and DWV loads, and the unique genetic background that characterizes each honeybee colony - due to the different origin of the queen and the males she mated with.”

Lines 161-163 “These colonies derive from an initial set of commercially acquired queens - used to establish the two apiaries - that were a cross of *Apis mellifera carnica* and *Apis mellifera mellifera*.”

As for the reference to the different geographic origin of the colonies, we are not sure what this refers to. The colonies were from two apiaries, true, but these apiaries are only 20 km apart, so we don't think the geographic range is so broad to justify significant differences. We have highlighted

this in the text and stressed that the two apiaries were chosen because of the different Varroa treatments, suitable to produce significant differences in viral loads that were fundamental for our study.

Lines 158-159: “The honey bee samples used in this study were obtained from two colonies located in two apiaries near Aberdeen, Scotland, 20 km apart from each other”

Lines 168-172: “We intentionally targeted colonies located in the two apiaries to obtain honey bee samples with expected significant variation in viral loads.”

b. The authors narrowed down the analysis to Colony A, but it is unclear whether the sample sizes for behavior (poor and good learners) and DMV load (low and high) provide adequate statistical power.

Thank you for raising this point that clearly requires clarification. Narrowing down to colony A was done only when assessing the effect of DWV loads on gene expression: this reduced the sample size to 16 honeybees in total (10 with high DWV and 6 with low DWV), which is smaller than the initial set, of course, but still suitable for transcriptomic analyses and well in line with other studies that have assessed similar research questions - for example, Pizzorno et al. (2021) used 12 bees in total. We understand your point though and we have added some considerations in the discussion:

Lines 479-483 “Perhaps the limited sample size (17 bees when considering Good Learners alone) could explain this outcome and a larger set of samples might increase the statistical power to detect significant differences; it must be noted though that the sample size we used is in line with previous studies in the same area (e.g., Pizzorno et al. 2021).”

As for the behavioural analyses, these were done instead on the whole set of 33 samples coming from both colonies. In this case therefore the comparison was between 17 good learners and 16 poor learners, which is well above the minimum recommended for transcriptomic studies and the average observed in other studies. We have added sample sizes in the main text of the results, where relevant, to clarify these aspects.

Also note that forager bees were collected within a one-hour window, factors such as age, foraging experience could still vary and impact both behaviors and viral load.

We agree with you that knowing the exact age of foragers would improve consistency of results and would remove noise in the transcriptomic signal due to age differences. Our success rate of reintroducing tagged bees to parental colonies to monitor them through foraging career has been extremely low, making it very challenging to incorporate this element in the experimental setup.

Lines 620-624 “Of note, we could not control for the age of the foragers that we sampled, another factor that has the potential to influence the transcriptomic response to a viral infection. Tagging individual bees and testing cohorts of foragers of comparable age is challenging but could surely be attempted in the future.”

c. Including a clear schematic of the experimental design would help readers.

Thank you for this suggestion. We have prepared a schematic that visualizes the details of the PER protocol, on one hand, and the full pipeline to obtain transcriptomic data from mushroom bodies samples, on the other hand. This is now **Figure 1**.

d. Provide high-resolution statistical figures with accessible and reader-friendly for better data visualization.

We have reformatted the figures to make labels and the reported statistical outputs more visible. In particular, we have moved panel labels at the top (and made them larger) in Figure 5, and we

have converted the supplementary figures reporting the outcome of correlation analyses to landscape format, so that we could use larger text for the labels.

In the differential gene expression analysis, clearly label comparison groups and ensure that axis labels are legible at publication size.

We have checked that all labels are clear in the pictures (including supplementary materials) and that font size is suitable for publication.

e. State a brief rationale at the beginning of each results subsection to frame the purpose of the analysis being presented. The discussion is too lengthy, contains repeated descriptions of results, and would benefit from moving some interpretive content to the result sections.

We agree that parts of the discussion were repetitive as you pointed out. We have therefore streamlined where possible and removed parts that were evidently redundant - see for example combining the first two paragraphs in a single introductory paragraph that provides an overview of the main results.

We have operated in the same way for the conclusions section, which appear much more compact now.

As for the results, we have followed the suggestion and added a short summary at the beginning of each section: **Lines 323-325** “We initially performed a series of analyses to explore the overall patterns of gene expression in the 33 mushroom body samples, and to enquire whether there was any factor in particular responsible for major differences in transcriptomic profiles.”

Lines 385-386 “In a third set of analyses we specifically focused on exploring the effect of harbouring a natural DWV infection on the transcriptomic profile of mushroom bodies.”

Lines 415-420 “To explore at a finer scale the possible interaction between DWV particles in the mushroom bodies and brain gene expression, we performed correlations analyses (Spearman’s rank correlation coefficient) where we plotted viral genome equivalents (expressed as log₁₀ transformation of total viral counts) against the levels of expression of individual genes (expressed as log₁₀ transformation of normalised read counts).”

Lines 446-449 “The set of gene expression analyses that we performed highlighted the presence of several lncRNAs in our list of differentially expressed genes. We decided therefore to explore the potential role of these molecules as regulators of gene expression by building gene regulatory networks for both datasets.”

Minor comments:

1. Replace honeybee in the title with *Apis mellifera* and expand DWV.

We have amended the title following these suggestions. The title now reads “Neurotranscriptomic profiling of Deformed Wing Virus-infected honey bee foragers (*Apis mellifera*) with different cognitive abilities”.

2. Italicize genus and species names throughout the manuscript.

We have proofread the manuscript again and addressed this where needed.

3. Expand all abbreviations at first use and ensure consistency thereafter.

We have proofread the manuscript again and addressed this where needed.

4. In keywords, consider using transcriptomics rather than neurotranscriptomics.

We have modified the keywords accordingly

5. Revise for overly long or complex sentences, and correct typographical errors - for example, remove log (line 954), bee (line 971), brackets for the reference (lines 181- 182), etc.

We have addressed these points in the text.

6. Provide references for statement in lines 101-104.

Reference added

7. Line 164, include units. AM?

Units included (AM)

8. Line 179, clarify how assays were performed in darkness.

Clarified (in a cabinet)

9. Specify the recommended room temperature for assays.

Clarified (22-25 C)

10. Provide catalog or product number for all reagents and instruments used in this study.

This isn't something we've typically done in the past and we're not sure it's necessary according to the journal guidelines. We also checked some recently-published manuscripts in Biology Open (e.g., <https://journals.biologists.com/bio/article/doi/10.1242/bio.061793/368964/Insulated-piggyBac-and-FRT-vectors-for-engineering>) and couldn't find evidence that would support including catalog or product numbers for reagents and equipment included.

11. Line 184, clearly state the concentration of citral odor used.

Concentration stated (95%, v/v)

12. Lines 191-193, assay steps are confusing, needs clarification.

We have rephrased and added a few more details and we hope that everything is clear now.

Lines 198-204 “Foragers that exhibited PER to the first citral exposure were discarded, as these bees were responding randomly to the stimulus at this point, since the conditioning phase had not yet taken place. If no PER was recorded for the odour itself during the first trial, the stimulus was reinforced with a 30% (w/w) sucrose solution, the reward. Individuals not responding to sucrose were discarded before further testing, as it would have been impossible to proceed with the conditioning phase with these individuals.”

13. Line 196, what is the author recommended time duration for freezing?

We are unsure about this query, as it is not clear whether it relates to a) time needed to sacrifice a bee via freezing? or b) recommended time (maximum?) to store samples in a freezer for molecular work?...In the first case, because the freezer was a -80 C, the process was super-quick (less than a minute for individual bees). Therefore, ideally, one can proceed with molecular work right after - but we waited a few months as normally there is very little time to perform molecular processing of samples when behavioural experiments are ongoing. If the query is more related to (b), samples are stable in a -80 C freezer and suitable for molecular work up to a few years after storage - assuming they have not been thawed (fully or partially) in the meantime.

We have added a sentence to clarify the timeline for molecular work.

14. Sample ID 8 is missing- clarify whether it was intentionally excluded or if this is an error in numbering.

Sample 8 was intentionally excluded - we have added a sentence in the legend of table to clarify this: “Note that sample #8 is missing due to insufficient yields obtained after RNA isolation.”

15. Report the sample numbers for each group in the figure legends.

We have added sample numbers to the relevant figures (2, 3, and 4).

Reviewer 2: This manuscript investigates two colonies of honeybee foragers looking for links among (a) natural levels of viral infections, (b) transcriptomic variations, and (c) cognitive abilities.

Towards this goal, they tested animals in an associative learning assay and then harvested their mushroom bodies to determine viral load and transcriptomic profile.

Curiously, the authors find little or no evidence for associations between cognitive ability vs. viral load or cognitive ability vs. changes in gene expression. This negative result runs counter to several prior studies done in laboratory honeybee strains. In contrast, their studies uncover several genes and RNA that correlate (negatively or positively) with viral load.

Several questions arise:

(1) In data where learning ability is a factor, only Poor learners are discussed (e.g. Table 3). Why are data from Good learners overlooked in these comparisons? Is it possible that correlations seen in Poor learners are reversed in Good learners? As the authors note, DWV has been shown to paradoxically improve and impair cognitive function in honeybees. As Good learners are not included in the main Tables, I am wondering if they have nothing to offer in this study?

We are sorry that our manuscript was not fully clear regarding the approach that we followed for these analyses. For Good learners we adopted exactly the same approach as for Poor learners but the statistical analyses revealed no difference in gene expression with the threshold that we adopted ($p=0.05$, pretty standard for these types of studies). This is stated at lines 349-352 of the manuscript and we have now added it to the legend of Table 3 for clarity. It is therefore not justified to proceed with additional analyses on this set of the data.

As for the correlation analyses, these were done on the two datasets that contained differentially expressed genes, which means the Poor Learner dataset and the Colony A dataset - this last one including both Poor and Good learners that were sampled from that colony. We have added a clarification at the start of the relevant paragraph in the results:

Lines 420-422 “Correlations analyses (Spearman’s rank correlation coefficient) between gene counts and viral loads present in the mushroom bodies were performed on all differentially expressed genes contained in the two datasets of interest, Poor Learners and Colony A: both datasets revealed some interesting patterns”

Along these lines, I am unsure if having "...cognitive abilities" in the manuscript is meaningful.

We believe that it is relevant to highlight that component of our study, starting from the title. The procedure of obtaining two groups of learners through a time consuming protocol is a significant component of the study, and is one of the novelty of this research. Brain gene expression in bees in response to DWV has been assessed before, but not in two groups of bees that clearly showed different abilities of solving a cognitive task. The fact that the transcriptomic difference between Good and Poor learners was weak, doesn’t justify removing this component from the paper in our opinion.

(2) The animals were collected during 9-10 am. Could the authors comment on the choice of the collection time? I am wondering how the time-of-day could be affecting cognitive ability, DWV load and transcription profiles, given the circadian clock controls much of honeybee physiology.

We agree that many genes can follow the insect circadian rhythm and respond accordingly, and we also agree that it would be interesting to explore what the extent of this response is - though this was certainly not the scope of such a preliminary study. We decided therefore to restrict the analysis to a specific time window to improve consistency in our observation and avoid that variation of time could cause background noise in the transcriptomic signal. As for the specific time selected, this was early morning to accommodate for the long time needed to prepare the samples for the first steps of the PER protocol: samples had to be collected in the morning to go through initial harnessing, sucrose and water sensitivity, conditioning and first memory test before the overnight pause that was required prior to the second memory test. Starting later in the day would have not provided enough time to accomplish the first steps within the same day. We have clarified this in the methods section:

Lines 172-174 “Honey bee foragers were collected in the summer of 2020 between 9.00 and 10.00 at the colony entrance when returning from a foraging trip.”

As for the possibility that DWV levels might change during the day following a circadian rhythm - if this is what you are implying - we admit we have not considered this as we are not aware of any studies that have addressed this possibility in this host-parasite system. Of course we cannot exclude that such a mechanism might be in place and we have added some considerations in the discussion.

Lines 614-624 “Third, we have considered a single time point post-infection, which cannot capture the probable varying temporal transcriptomic responses to DWV infection: not only these responses might vary progressively over time, as a result of changing physiological conditions, but also in a

cyclical fashion, as a consequence of natural circadian rhythms that are known to play a key role in the overall regulation of gene expression in insects (Sandrelli et al. 2008).”

(3) Since most of the paper focuses on Colony A, it would be helpful to have PCA analysis done on just Colony A data. The analysis could be drastically different because without Colony B, the data have much less variance. I wonder how the single colony data would cluster.

This is an excellent suggestion, thank you for bringing this up. We had in fact produced additional PCA analyses, to explore data distribution in different groups of bees. In general no striking patterns appear, and therefore we had decided to not show these results. However, following up on your point, we are now including these outputs in the supplementary materials (Figure S6). When considering colony A alone (Newburgh), there is slightly more separation according to DWV loads, but the clustering is not clear-cut and the PC loadings are generally low.

(4) Since the natural DWV load has no clear effect on learning as assessed through PER, could the authors speculate on what type of effect on cognition might these natural levels of virus have in these bees? For example, if an assay measured collective cognition would Colony A fare worse than Colony B insects?

Thank you very much for raising this point, which prompted us to consider possible effects of DWV infection at the colony level, like for example social learning or communication. We have added a few considerations in this direction in the discussion:

Lines 515-523 “In line with these considerations, we cannot exclude that the presence of DWV infections in the mushroom bodies might result in the modification of other types of learning, for example social learning, that are widespread in honey bees and other social insects, rely on complex systems of communications and have a transcriptomic basis (e.g., Veiner et al. 2022; Manfredini et al. 2023). As a matter of fact, DWV and other honey bee viruses have been shown to interfere with chemical recognition and odour perception (e.g., Geffre et al. 2020; Silva et al. 2025), and it will be interesting in the future, for example, to investigate whether such viruses have any effect on the honey bee waggle dance communication.”

Minor comments:

(1) Figures 2 and 4 missing x-axis label.

Axes added

(2) Figure 5: Would be clearer if panel titles such as “(A) F-box/LRR...” appeared above each scatter plot instead of below.

The labels are now above the panels

(3) Line 422: “Both the Colony A and Poor...” What do the authors mean by scale-free topology and why is that significant?

There is now a description of scale-free-topology and why it is relevant, **lines 454-460** “Both the Colony A and Poor Learner GRNs conformed to a scale-free topology (Kolmogorov-Smirnov $p \geq 0.99999$; $\alpha \approx 2.6-2.7$), meaning that their degree distributions follow a power law. In such networks, most nodes have few connections, while a small number of hubs have many. This property is characteristic of biological networks and supports the biological plausibility of the identified lncRNA hubs and regulator-target interactions.”

(4) The current Figure 1 shows that gene expression data cluster in terms of colony of origin and Figure 3 shows that Colony A gene expression has more uniformity. To me, these two data belong together, and might be helpful if they appeared in the same figure or one right after the other (if in separate figures). Right after presenting these data, the authors would focus on Colony A data for the rest of the paper. Why not get these out of the way in the beginning?

Thank you for raising this point. In reality we still use both colonies for the poor learner analyses that follow the presentation of PCA outputs. However, we like the suggestion of presenting the two figures together at the beginning of the result section and we have therefore moved the heatmap up (together with the relative text in the manuscript).

(5) Figure S3: For each panel, please enlarge font size for the gene ID and corresponding statistics. Currently, they are hard to read.

We have increased the image size by changing the relevant pages to landscape orientation and splitting across several pages where necessary.

We have addressed this in response to the comments above. In particular, we have rephrased the title but we have decided to retain the reference to bees' cognitive abilities as explained in response to one of the comments of Reviewer 2.

Second decision letter

MS ID#: bio.062204R1

MS Title: Neurotranscriptomic profiling of Deformed Wing Virus-infected honey bee foragers (*Apis mellifera*) with different cognitive abilities

Authors: Simon E. Loughran; Lauren Dingle; Alan S. Bowman; Fabio Manfredini

I've had the time this morning to fully read through your responses to the Reviewers' comments and your associated manuscript edits, and I am happy to tell you that your manuscript has now been accepted for publication in Biology Open, pending our standard publication integrity checks. It was accepted on 21st October 2025.